# Investigating the effect of clinical pharmacist intervention in transitions of care on drug-related hospital readmissions among the elderly: study protocol for a randomised controlled trial

Jonas Kindstedt [1], Sofia Svahn [1], Maria Sjölander [1], Eva-Lotta Glader [2], Hugo Lövheim [3], Maria Gustafsson [1]

¹Department of Integrative Medical Biology, Umeå University, Umeå, Sweden
²Department of Public Health and Clinical Medicine, Umeå University, Umeå, Sweden
³Department of Community Medicine and Rehabilitation, Umeå University, Umeå, Sweden

**Correspondence to**
Mr Jonas Kindstedt;
jonas.kindstedt@umu.se

## ABSTRACT

**Introduction** Drug-related problems (DRPs) are a major cause of unplanned hospital admissions among elderly people, and transitions of care have been emphasised as a key area for improving patient safety. We have designed a complex clinical pharmacist intervention that targets people ≥75 years of age undergoing transitions of care from hospital to home and primary care. The main objective is to investigate if the intervention can reduce the risk of unplanned drug-related readmission within the first 180 days after the person is discharged from hospital.

**Methods and analysis** This is a randomised, controlled, superiority trial with two parallel arms. A total of 700 people ≥75 years will be assigned to either intervention or routine care (control). The intervention, which aims to find and manage DRPs, is initiated within a week of the person being discharged from hospital and combines repeated medical chart reviews, phone interviews and in some cases medication reviews. People in both study arms may have been the subject of a medication review during their ward stay. As the primary outcome, we will measure time until unplanned drug-related readmission within 180 days of leaving hospital and use log rank tests and Cox proportional hazard models to analyse differences between the groups. Further investigations of subgroup effects and adjustments of the regression models will be based on heart failure and cognitive impairment as prognostic factors.

**Ethics and dissemination** The study has been approved by the Regional Ethical Review Board in Umeå (registration numbers 2017-69-31M, 2018-83-32M and 2018-254-32M). We intend to publish the results with open access in international peer-reviewed journals and present our findings at international conferences. The trial is expected to result in more than one published article and form part of two PhD theses.

**Trial registration number** NCT03671629

## INTRODUCTION

An ageing population with multiple morbidities and medications brings with it considerable challenges for healthcare and society

---

**Strengths and limitations of this study**

► One of few multifaceted clinical pharmacist interventions directed at transitions of care.
► Targets elderly people ≥75 years of age, a vulnerable group rarely addressed in clinical research.
► Conducted in a setting where medication reviews by clinical pharmacists are already implemented during routine hospital care.
► The participants are not blinded to the treatment assignments.

---

in terms of adverse drug reactions and other drug-related problems (DRPs).[1] Hospital admission rates increase with age and,[2–4] although prevalence estimates differ between studies,[5] it has been observed that up to 30% of unplanned hospitalisations among the elderly are caused by DRPs,[6 7] a percentage that appears to be even higher in the vulnerable subgroup of old people with major neuro-cognitive disorder (NCD).[8] The transition of care between different healthcare providers has been associated with a high risk of inappropriate prescribing, medication errors and adverse drug events,[9–11] and the World Health Organisation (WHO) has identified transitions of care as a key issue for improving patient safety in primary healthcare.[12]

Poor medication adherence is a major concern in developed countries given that adherence rates average 50% among people with long-term therapies.[13] The mechanisms of adherence are multifactorial and, even though there is little evidence to support old age alone as an independent predictor of non-adherent behaviour, the elderly are still at higher risk due to multiple morbid conditions and polypharmacy.[14] Importantly, it has also been observed that even milder forms

**Table 1** Important aspects of the medication reviews conducted by clinical pharmacists employed by Region Västerbotten

| | |
|---|---|
| Medication reconciliation | The pharmacist ensures that the medication records are updated and accurate. This assessment is based on both hospital and primary care medical records |
| Overall review of list of medications | This includes indications for therapies, correct choice of drugs, dosages, treatment durations and untreated indications |
| Clinical symptoms in relation to drug treatment | Symptoms of ADRs |
| Impaired body function | Liver and renal function, swallowing difficulties, contraindications and allergies |
| Drugs that require specific attention | Toxic drugs, drugs commonly associated with side effects and PIMs |
| Interactions | Drug–drug interactions and drug–food interactions |

ADR, adverse drug reaction; PIM, potentially inappropriate medication.

of cognitive impairment severely affect adherence to medication.[15] In the context of transitions of care, poor adherence in the sensitive postdischarge phase has been associated with hospital readmission and appears to be of particular clinical relevance among people suffering heart failure.[16–18]

The clinical pharmacist is one of many professions involved in preventing and managing DRPs. In principle, clinical pharmacy can be described as the science and practice of rational medication use and includes both medication reviews and counselling with patients as well as other healthcare professionals.[19] Medication reviews conducted by clinical pharmacists have demonstrated a positive overall effect on both the appropriateness and cost of medication.[20 21] Regarding the effect on hospital readmissions and other clinical outcomes, review data are less consistent and incorporate few randomised controlled trials (RCTs)[22–24]; however, there are relatively recent findings indicating that medication reviews conducted by clinical pharmacists on the ward can reduce drug-related readmissions to hospital in various groups of elderly people.[25–27] Postdischarge interventions involving phone-based follow-ups by clinical pharmacists have reduced all-cause readmission rates, but these trials did not specifically target an elderly population.[28 29] In addition, there seems to be a need for more RCTs that target medication discrepancies and errors during care transitions between hospital and primary care.[30] In summary, there is still insufficient evidence for an optimal postdischarge intervention model that incorporates adherence elements and specifically addresses transitions of care involving elderly people.

In the present study, our primary aim is to investigate the effect of clinical pharmacists in transitional care on unplanned drug-related readmissions among people ≥75 years of age in a context where medication reviews conducted by clinical pharmacists within the hospital are already common practice. Our primary objective is to assess if intervention by a clinical pharmacist during a transition of care can reduce the risk of unplanned drug-related hospital readmission for 180 days after the person leaves the hospital. As secondary objectives, we will study the effects of the intervention on unplanned drug-related readmissions within 30 days, unplanned hospital visits (all-cause readmissions and emergency department visits), mortality, medication adherence and quality of life. Subgroup analyses will be based on the occurrence of heart failure and cognitive impairment. Moreover, we will collect qualitative interview data regarding the intervention.

## METHODS AND ANALYSIS
### Principal study design
This study is conducted as a randomised, controlled, superiority trial with two parallel groups. The control group will be assigned to routine care while the intervention group will receive an extended clinical pharmacist service for a period of 180 days, a procedure that incorporates regular medical chart reviews, medication interviews, medication reviews (table 1) and collaboration with the primary care physician to address DRPs. It should be noted that study participants in both groups may already have been the subject of a medication review performed by a clinical pharmacist during their hospital stay.

### Study population, recruitment and randomisation
The recruitment of study participants started in September 2019 and takes place among emergency admissions to the University Hospital of Umeå. We will continuously recruit participants until we reach the target sample size of 700 randomised individuals. Thus far, recruitment has been limited to one medical ward; however, it is possible

**Box 1   List of eligibility criteria**

Inclusion criteria
► ≥75 years.
► Living at home (ie not in nursing home).
► Emergency admission.
► Registered at one of nine specified primary care centres.
Exclusion criteria
► Do not speak Swedish or unable to communicate.
► Admitted due to intoxication (drug or alcohol).
► Scheduled for palliative care.

to increase the number of wards if the inclusion rate proves insufficient to reach the target sample size during 2022. People who meet the eligibility criteria are given the opportunity to participate in the trial (box 1). Once written consent has been obtained (approval from next of kin in cases of people with major NCD), the participants report self-assessed adherence and quality of life through questionnaires. Following discharge from the ward, the participants are randomly assigned to one of the two study arms through a stratified randomisation procedure (figure 1) based on their results from a shortened 4-item version of the Gottfries' Cognitive Scale,[31] a validated tool for proxy rating cognitive impairment.[32 33] Within each stratum, the participants are further assigned to either of the study arms with an intended ratio of 1:1 through random allocation.

### Allocation concealment and blinding

We enrol participants before they are discharged from hospital. The allocation sequences are computer

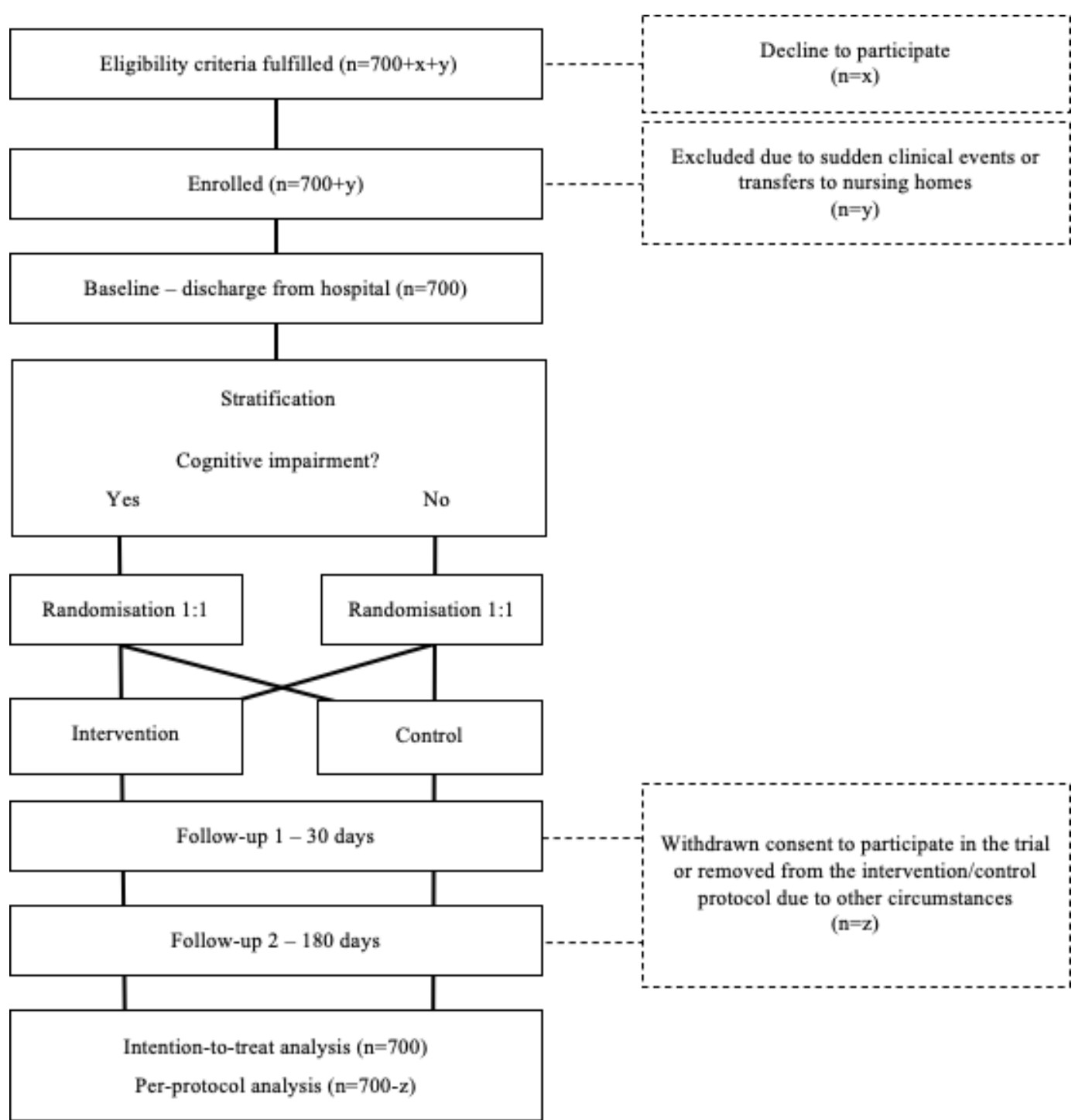

**Figure 1** Study flow chart from recruitment to final follow-up.

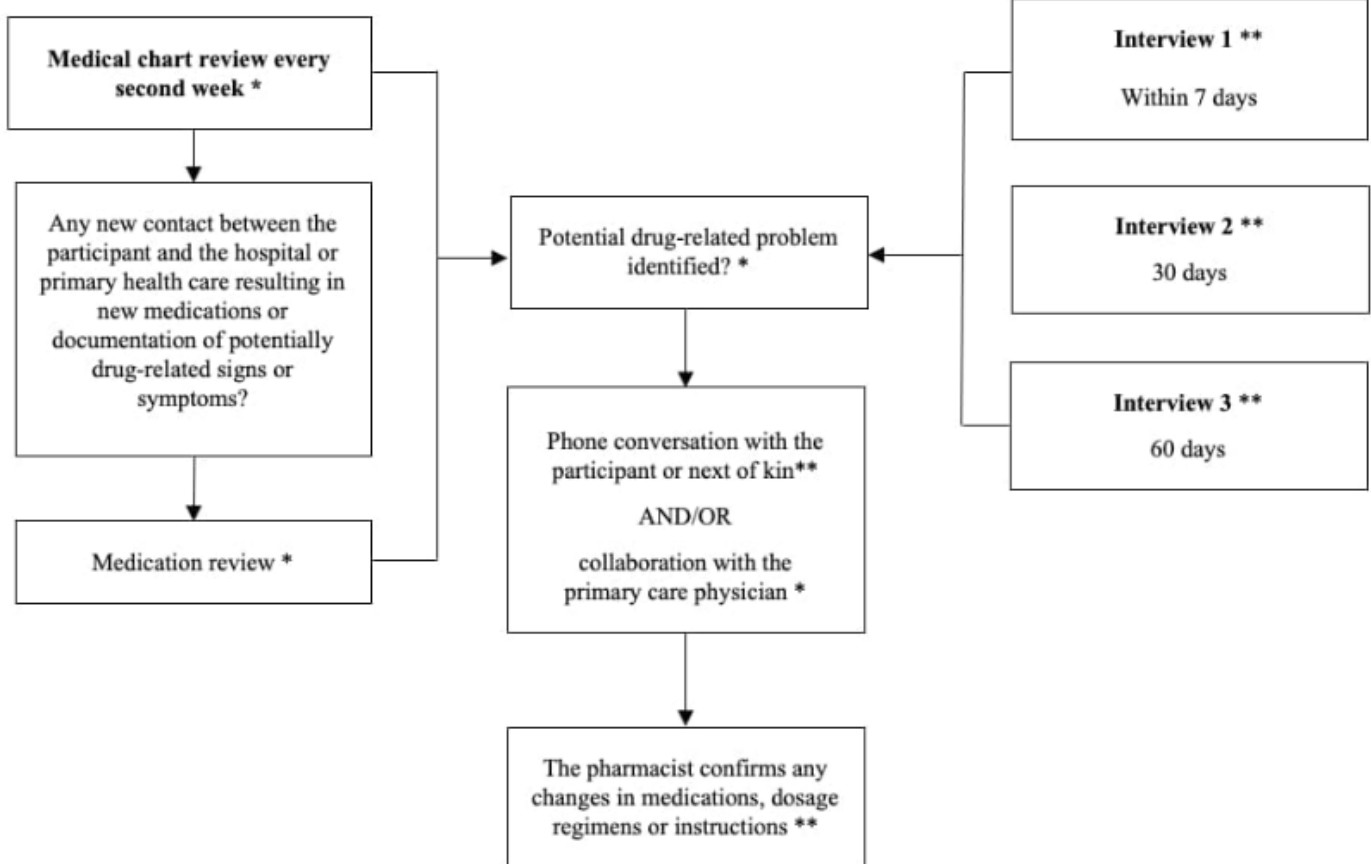

**Figure 2** The key features of the intervention, including both prescheduled phone interviews and repeated medical chart reviews. The intervention is initiated immediately after hospital discharge and goes on for 180 days. *Clinical pharmacist activity. **Interactions between clinical pharmacists and study participants.

generated and managed by external administrators who irreversibly assign the participants to their respective treatments. From this stage onwards, the clinical pharmacists are aware of the group assignments. Study participants are considered blinded until the first phone call. The statistical analysis will be performed in agreement with a statistician who is unaware of the assigned treatments. All information about group assignments will also be unknown to the outcome assessors and any notes in medical records that might offer such clues will be removed beforehand.

### Intervention group protocol

Our intervention model (figure 2) incorporates both phone-based interaction with the participants and access to both hospital and primary care medical records. A Master of Pharmacy or similar qualification is required for all clinical pharmacists performing the intervention. The clinical pharmacist initially contacts the participants within 7 days of their discharge from hospital. Other phone-based interviews are scheduled at 30 and 60 days after hospital discharge. Due to practical considerations, interviews may be rearranged to the nearest possible date. In brief, these conversations focus on misunderstandings regarding new and ongoing treatments, adherence to

medication and other potential DRPs, ranging from side effects to practical administration barriers.

In addition to the prescheduled interviews, the intervention includes repeated medical chart reviews performed on average every second week during a period of 180 days after the participants have been discharged from hospital. This activity is implemented in order to identify discrepancies between the lists of medications before, during and after hospitalisation, unanswered referrals between healthcare providers, deviating laboratory values and whether the participant has had any relevant contacts with the healthcare system (figure 2). Moreover, the medical chart reviews also function as an important basis for the interviews. In the event of multiple medication alterations or any documented signs or symptoms that might be drug-related, the clinical pharmacist conducts a medication review (table 1). This process includes several steps, such as medication reconciliation, evaluation of proper dosage and indication, signs of ADRs or side effects, check for interactions and special considerations due to impaired renal function. In brief, the exact types of DRPs can differ considerably between participants and range from administration issues and poor medication adherence to drug-induced electrolyte imbalance or orthostatic hypotension. Identified or suspected DRPs that are

detected through the interviews or medical chart reviews but cannot be solved by direct communication with the participant or next of kin are instead communicated over phone with the primary care physician. In this dialogue, the clinical pharmacist describes the potential DRP being present and offers a recommendation on how to manage the issue. The physician is the person who ultimately decides on any changes to medication regimens, in which case the clinical pharmacist contacts the patient within a few days in order to confirm and reinforce the physician's instructions. All clinical pharmacists involved in the trial have electronic access to the participants' medical charts. This access includes physicians' notes, laboratory records, current lists of medications and epicrises. The combined information from these sources provide a basis for both medical chart reviews and medication reviews. During the intervention, the clinical pharmacist continuously documents the time spent on the different activities for each participant.

## Public and patient involvement

There are no elements of public and patient involvement in the design and conduct of this trial, and we have no current plans of involving patients or the public in the dissemination of the research findings.

## Data collection and outcomes

All outcome data will be gathered from medical charts with the exception of self-report questionnaires. All collected research data will be coded and kept in locked archives. Before, during and after the trial, all forms and protocols used during the interventions and follow-ups that reveal individual personal data are stored according to the same principle. There will be no identifiable data in the dissemination of the results. In summary, outcomes will be measured at the time of discharge from hospital (baseline) and then after 30 and 180 days, respectively (table 2). The follow-ups on Medication Adherence Report Scale (MARS-5) and EuroQuol five-dimension (EQ-5D) are conducted by phone and preferably in conjunction with the intervention phone calls. Rehabilitation needs require that some enrolled participants are transferred to the geriatric department before leaving the hospital for home. If a participant is relocated to another clinical ward, the actual discharge from hospital will be considered as baseline, with the exception of self-reported adherence and quality of life, which are reported on the original ward. As a consequence, we contact all study participants by phone at 30 and 180 days after their first report regarding MARS-5 and EQ-5D, even if this date differs from the actual date on which they were discharged from hospital.

## Hospital readmissions

The primary outcome is time until unplanned drug-related readmission within 180 days of being discharged from the hospital. As a secondary outcome, we also study these events within a narrower time frame of 30 days. The

**Table 2** Scheduled data collection for primary and secondary outcomes

|  | Baseline | 30 days | 180 days |
|---|---|---|---|
| **Clinical outcomes** | | | |
| Drug-related readmissions | | x | x |
| All-cause readmissions | | x | x |
| Emergency department visits | | x | x |
| Mortality | | x | x |
| **Adherence** | | | |
| MARS-5 | x | x | x |
| **Quality of life** | | | |
| EQ-5D | x | x | x |
| **Patient characteristics** | | | |
| Age and sex | x | | |
| Cohabitant and home healthcare | x | | |
| Diagnoses | x | | |
| No of medications | x | | |

Baseline data are collected from the date of hospital discharge. The data for MARS-5 and EQ-5D can be retrieved before the participants are discharged and the date on which questionnaires are completed are considered as baseline regarding these two outcomes.
EQ-5D, EuroQuol five-dimension; MARS-5, Medication Adherence Report Scale.

cause of each readmission will be retrospectively determined by a group of blinded physicians and pharmacists who are not otherwise involved in the study. This assessment group will reach a consensus decision regarding the causality of each readmission according to criteria from WHO, a method that has been used previously in similar studies.[25 34 35] Readmissions classified as certain, probable or possible in this assessment will be regarded as drug related in the following statistical analysis. We also measure time until unplanned hospital visit (all-cause readmissions and emergency department visits), time until death, as well as frequencies of unplanned drug-related hospital readmissions and unplanned hospital visits.

## Medication adherence

We measure study participants' self-reported medication adherence using the MARS-5, a self-assessment questionnaire with five items on a five-point scale that has previously been translated into Swedish.[36 37]

## Quality of life and health economics

Quality of life is reported through a Swedish translation of the EQ-5D questionnaire, a standardised instrument consisting of two different components.[38] In the first part, the respondents describe their health status in five different dimensions: walking ability, self-care, usual activities, pain/discomfort and anxiety/depressive symptoms.

This part is followed by a visual analogue scale on which the respondents evaluate their own health status on a scale of 0–100.

### Baseline characteristics

Baseline data include various personal characteristics, such as age, sex, living conditions in terms of cohabitation and home healthcare, diagnoses and current list of medications.

### Sustained use and system-wide implementation

In addition to quantitative data, there will also be a qualitative component of the study. In order to capture the functionalities of the model, healthcare staff and participants involved in the intervention, and in some cases their next of kin, will be interviewed. An interview guide will be used and the questions will mainly focus on what does and does not work, overall satisfaction and personal experiences.[39] The interview guide will also provide the opportunity to ask additional questions and elaborate on other issues. Moreover, the interviews will explore potential improvements, refinements to support 'non-adopters' and future implementation, as well as the opportunity to offer other comments and suggestions. The interviews will be audio recorded and transcribed verbatim, and the analysis will be conducted in a systematic process for developing codes and themes in interview data. While coding entails the work of naming the different categories, the thematisation process involves moving from so-called free-floating codes into an integrated framework of empirical themes.[40]

### Discontinuations and deviations from the protocol

An enrolled study participant can at any time withdraw consent to participate in the study or simply request that remaining phone calls be cancelled. Furthermore, the intervention can be discontinued for a number of other reasons, such as a change of living circumstances from home to residential care or repeated unanswered phonecalls. Importantly, we do not continue to record any new information from participants who withdraw their consent to participate in the study. Any documented death or withdrawn consent is censored in the survival analysis from that point and onwards. In all other cases of discontinued interventions, we collect data regarding hospital readmissions and emergency department visits for the complete 180-day period. Outcomes will primarily be analysed according to an intention-to-treat principle that includes all randomised participants for the time they were under observation. For outcomes other than time to event, we will apply the last observation carried forward in cases of missing data. Finally, there will be a supplementary per-protocol analysis of drug-related readmissions and questionnaires among participants who have received all prescheduled interviews during the follow-up period.

### Statistics

For a relative risk reduction of 40% in drug-related hospital readmissions from 19% to 11%, a total study population of 700 individuals is required to achieve a statistical power of 80% at our prespecified significance level of 0.05 for two-sided tests. The estimated incidence of drug-related readmissions was observed in a previous trial within the same community, although that study was limited to elderly people with major NCD.[25]

We will present time-to-event data in Kaplan-Meier survival curves and analyse differences in survival data between the groups through log rank tests and Cox proportional hazard models. Further investigations of subgroup effects and adjustments of the regression models will be based on heart failure and cognitive impairment as prognostic factors. We will analyse secondary outcomes using various appropriate statistical tests, such as $X^2$ and independent sample t-test, depending on the type of variable.

### Limitations and risk of bias

The first and second author of this protocol are involved in the trial both as researchers and as clinical pharmacists conducting the intervention. As previously mentioned, we will undertake several measures to reduce the risk of bias that could emerge from these multiple roles. For example, we will consult external staff for the primary outcome assessment and statistical analyses. Other clinical outcomes in terms of all-cause readmissions, emergency department visits and mortality are electronically documented data with no room for subjectivity. Therefore, this risk of bias mainly applies to MARS-5 and EQ-5D, which should be considered in the interpretation of our findings. Moreover, all interviews within the qualitative part of the study will be conducted by another person than the clinical pharmacist who performed the interventions on those specific participants. Due to the nature of the intervention, neither participants nor clinical pharmacists are blinded. We do not consider the lack of blinding to be a major issue for the study, but bias related to special attention and the knowledge of being observed could possibly affect self-reported adherence and quality of life among the participants.

## ETHICS AND DISSEMINATION

Prior to hospital discharge, people who fulfil the inclusion criteria are given oral and written information about the study on the clinical ward. Anyone who does not wish to participate is free to decline or withdraw at any time during the course of the trial. Individuals with major NCD participate without formal written informed consent, although in these cases both the intended participant and the next of kin are informed about the study and given the opportunity to decline participation. The study has been approved by the Regional Ethical Review Board in Umeå (registration numbers 2017-69-31M, 2018-83-32M and 2018-254-32M). No participant in this trial is subject to any clinical procedures that are likely to be harmful and there are therefore no routines for collecting, assessing, reporting or managing adverse events or other unintended effects of the intervention. Moreover, we

assume that any participants who are uncomfortable with the trial will exercise their right to withdraw from the protocols. There are no agreements or other regulations that will limit our access to collected data. The study will be reported according to the Consolidated Standards of Reporting Trials guidelines.[41] Our intention is to publish the results with open access in international peer-reviewed journals. We also aim to present the results of the trial at both national and international conferences. The trial is expected to result in more than one published article and form part of two PhD theses. In accordance with the guidelines issued by the International Committee of Medical Journal Editors,[42] all listed authors have been involved in developing or revising the study design, contributed in the review process of this article, approved the final version and will ensure that issues related to the accuracy or integrity of the work are investigated and resolved.

## Current trial status

The recruitment and enrollment of participants started in September 2018 and there were 20 people enrolled in the trial by the end of that year. From January 2019, additional staff were involved in the recruitment process and the inclusion rate has since increased. The collection of data is currently ongoing, and 161 people had been randomised by November 2019. The intervention, or phone-based follow-ups in the case of controls, had thus far been discontinued for 44 of these individuals. The main reasons were withdrawn consent (n=13), transfers to nursing homes (n=12), repeated unanswered phone calls (n=9) and death (n=5).

**Contributors** JK wrote this study protocol. MG, HL, E-LG and SS developed the original study design. JK, SS, MS, E-LG, HL and MG were all involved in the revision of the study design and contributed in the review process of the protocol manuscript. JK and SS are jointly responsible for the collection of data and administration of study participants. JK and SS are also involved in the trial as two of the clinical pharmacists performing the intervention. MG is the principal investigator and responsible for the funding and overall management of the trial.

**Funding** This study is supported financially by grants from Region Västerbotten, the Swedish Society of Medicine, Umeå University Foundation for Medical Research, Forte (ref: 2017-01438) and the Swedish Research Council (ref: 2019-01078).

**Disclaimer** The funders have no role in the study design, data collection and analysis, decision to publish or preparation of the manuscript.

**Patient and public involvement** Patients and/or the public were not involved in the design, conduct, reporting or dissemination plans of this research.

**Patient consent for publication** Not required.

**Provenance and peer review** Not commissioned; externally peer reviewed.

**ORCID iDs**
Jonas Kindstedt http://orcid.org/0000-0002-9422-5125
Sofia Svahn http://orcid.org/0000-0001-5229-5988
Maria Sjölander http://orcid.org/0000-0002-8364-6290
Eva-Lotta Glader http://orcid.org/0000-0003-4095-6501
Hugo Lövheim http://orcid.org/0000-0002-5271-4780

Maria Gustafsson http://orcid.org/0000-0003-3615-4880

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
