## [Reviewer comments · BMJ Open]

ARTICLE DETAILS

TITLE (PROVISIONAL)	STUDY PROTOCOL FOR A RANDOMISED CONTROLLED TRIAL INVESTIGATING THE EFFECT OF CLINICAL PHARMACIST INTERVENTION IN TRANSITIONS OF CARE ON DRUG-RELATED HOSPITAL READMISSIONS AMONG THE ELDERLY
AUTHORS	Kindstedt, Jonas; Svahn, Sofia; Sjölander, Maria; Glader, E-L; Lövheim, Hugo; Gustafsson, Maria

VERSION 1 – REVIEW

REVIEWER	Prof. Nicolas Rodondi Inselspital Bern, Switzerland
REVIEW RETURNED	17-Jan-2020

GENERAL COMMENTS	Methods 1. It hasn't been decided yet if the primary outcome (drug-related hospital readmissions) will be adjudicated or assessed with another method. This should be fixed in the protocol as it may introduce a bias depending on which option is chosen.2. The criteria for participant exclusion during the study needs to be well defined and patients should not be excluded if the assessment of the primary endpoint is possible, even though they didn't respond to phone calls. Currently patients can be excluded from the study (and 9 already have) for not answering phone calls, however it isn't clear if endpoint assessment or intervention phone calls are meant. How will they be accounted for in the analysis?3. As the intervention group will undergo 3 different interventions (review of the medical chart, medication review and medication interview), why do you exclude patients for not doing the medication interviews when the other 2 interventions can still be done?4. As the primary outcome (drug related readmissions) will be assessed through the medical record and not patient-reported, excluding patients for not answering phone calls (so not following the intervention) does not result in an intention-to-treat analysis as is written in the protocol and is an important risk of bias.5. It says that the intervention will last 180 days, but the planned phone-based interviews are at day 30 and 60. Does the rest of the intervention (medical chart review every 2 weeks) go on for 180 days? If yes please specify in the description of the intervention.6. Please describe the intervention with more detail in the text: what will the medication review look at? Which kind of drug related problem? If a drug is found to be inappropriate, what would be done exactly with the patient? How will the communication with the primary care physician take place and what will be discussed7. Blinding: did this study try to do partial blinding of participants?
--

	8. Is it planned to account for clustering by physician in the analysis plan. 9. Please specify where and how you will get the information for the medical chart review and medication review. 10. The planned subgroup analysis for heart failure and cognitive impairment makes sense as you expect a difference in effect of the intervention in those groups based on other studies. However, as this is a randomized study and the groups are even stratified for cognitive impairment, you do not expect a rate difference between the intervention and control arms and thus no adjustment of the regression model should be necessary. Abstract 1. The intervention should be better described in the summary 2. Point for the subgroup analysis applies to the abstract also.
--	--

REVIEWER	Ulrich Jaehde Institute of Pharmacy University of Bonn Germany
REVIEW RETURNED	26-Jan-2020

GENERAL COMMENTS	The manuscript describes the protocol for a randomised controlled trial investigating the effect of a clinical pharmacist intervention in elderly patients after discharge from the hospital. The intervention consists of three telephone calls and repeated medical chart reviews by the clinical pharmacist. As there is a strong need for specific interventions assuring patient safety in transitions of care, the study results will be of great interest for health care providers and quality managers striving for a safer medication process after discharge. The study is well recruiting. From September 2018 to November 2019 161 patients were randomised. The manuscript is well written, the methods are state-of-the-art and the protocol is described clearly and presented in the context of the literature. The protocol addresses the items of the SPIRIT checklist where appropriate. Therefore, the manuscript is acceptable for publication after consideration of the following minor issues: Is the time that the clinical pharmacist needs for the intervention documented? If yes, this should be mentioned as it will be an important information for potential future financing of the intervention. On page 9 the authors state that a medication review is conducted in the event of new medications or any documented drug-related symptom. This information should be added to the scheme in Figure 2. Page 10: It is not clear if the outcome measurements after 30 and 180 days are performed orally or using a questionnaire. If it is done by a phone interview is it after 30 days done in conjunction with the phone call which is part of the intervention? Page 11: The assessment of the causality of readmission is a major issue but remains obscure. The authors state that they will conduct a review of available options in order to find a valid and feasible classification tool. Since 16 months have passed after initiation of the study: has this review process been finished in the meantime? If yes, what is the result? If not, please give more details on the available options. Page 12: The qualitative part of the study should be explained in more detail, especially regarding the methodology.
--

	Authors' contributions: The first author of the manuscript (assuming that he is one of the PhD students) is also one of the clinical pharmacists performing the intervention. Thus, he is in a double role and in fact evaluating his own performance. The authors should comment if this "personal union" can cause a major bias in the study results which should be considered when interpreting the results.
--	---

VERSION 1 – AUTHOR RESPONSE

Reviewer 1:

Reviewer Name: Prof. Nicolas Rodondi

Institution and Country: Inselspital Bern, Switzerland

Competing interests or state: None declared

This is a study protocol for a randomized controlled superiority trial on the impact of a clinical pharmacist intervention in transition of care on unplanned drug-related hospital readmissions at 180 days in elderly patients. Other outcomes will be unplanned hospital readmissions within 30 days, unplanned hospital visits, mortality, medication adherence and quality of life. The study will take place at the University Hospital of Umea and started in September 2019.

The research question is relevant as there is a need for more trials concerning optimization of post-discharge care for elderly people. The study design is appropriate to answer the research question and the outcome are well chosen.

1. It hasn't been decided yet if the primary outcome (drug-related hospital readmissions) will be adjudicated or assessed with another method. This should be fixed in the protocol as it may introduce a bias depending on which option is chosen.

Thank you for highlighting the importance of being clear on this point. For the sake of transparency, we have decided to adhere to the original plan i.e. consult an external expert group of health care professionals to judge the causality of each hospital readmission. This is now specified in the revised protocol on page 9. To our knowledge, this appear to be the most reliable alternative.

2. The criteria for participant exclusion during the study needs to be well defined and patients should not be excluded if the assessment of the primary endpoint is possible, even though they didn't respond to phone calls. Currently patients can be excluded from the study (and 9 already have) for not answering phone calls, however it isn't clear if endpoint assessment or intervention phone calls are meant. How will they be accounted for in the analysis?

Thank you, we realize that our previous description of discontinuations could be interpreted in different ways. We do not exclude participants from the endpoint assessment for not responding to phone-calls. These individuals are only excluded from future intervention. Of course, we want our endpoint assessment to be clearly described and we have added clarifications regarding this issue on page 11.

3. As the intervention group will undergo 3 different interventions (review of the medical chart, medication review and medication interview), why do you exclude patients for not doing the medication interviews when the other 2 interventions can still be done?

This is a valid observation and the medical chart reviews could theoretically go on for the rest of the 180 day-period. However, we consider the different elements of the intervention importantly connected to each other. The medical chart reviews serve as a basis for the phone calls during the first 60 days of the intervention. In addition, most of the potential DRPs that are suspected during the medical chart reviews are generally discussed and verified with the participant rather than the physician. We have added some information about this on page 2 and 7 in the revised version of the manuscript.

4. As the primary outcome (drug related readmissions) will be assessed through the medical record and not patient-reported, excluding patients for not answering phone calls (so not following the intervention) does not result in an intention-to-treat analysis as is written in the protocol and is an important risk of bias.

We think this matter relates to comment No. 2. As mentioned, we only exclude these participants from future intervention, not the outcome assessment. Missing data is only an issue in cases of withdrawn consent to participate in the study. Due to ethical considerations, we will only extract data until the date of the withdrawn consent. Consequently, all participants will still provide time-to-event data for the time they were under observation, and we will apply "last observation carried forward" for the other outcomes to account for missing data in the intention-to-treat analysis. We have revised the section on page 11 to avoid confusion regarding exclusions, withdrawals, and discontinued interventions.

5. It says that the intervention will last 180 days, but the planned phone-based interviews are at day 30 and 60. Does the rest of the intervention (medical chart review every 2 weeks) go on for 180 days? If yes please specify in the description of the intervention.

Thank you for noticing this lack of information, which has now been added on page 7.

6. Please describe the intervention with more detail in the text: what will the medication review look at? Which kind of drug related problem? If a drug is found to be inappropriate, what would be done exactly with the patient? How will the communication with the primary care physician take place and what will be discussed

This is a difficult step to describe in the protocol since it depends on the type of potential DRP being present, and to some extent we rely on the clinical pharmacist's own knowledge and judgement. Nevertheless, we have added a few general examples of typical DRPs on page 7. The aspects of the medication review are summarized in Table 1 on page 5, which applies to medication reviews both during ward stay and post-discharge. Some of the main features of medication review have also been mentioned in the text for a more detailed description. Primary care physicians are contacted by phone and we have now mentioned this procedure on page 7.

7. Blinding: did this study try to do partial blinding of participants?

This is a very good and interesting observation. It could probably be considered a type of partial blinding as we do not explicitly tell the participants whether they have been assigned to intervention or control. The intervention group should obviously realize that they are receiving the intervention, but we think the majority of the controls do not distinguish between the intervention and the phone-based 30-day follow-up regarding EQ-5D and MARS-5. This is relevant since we want to investigate the effect of the clinical pharmacist without the influence of the actual phone call. As a consequence, we think the phone interaction with both groups may actually reduce potential bias related to special attention and observer effects. Still, all participants are informed that they are being part of a randomized trial in the information letter provided during recruitment, and anyone who asks will be informed about their treatment assignment. Therefore we chose not to label it as partial blinding in the manuscript. The difficulty to perform this type of intervention with fully blinded participants has now been added as a limitation on page 2, and is further discussed on page 12.

8. Is it planned to account for clustering by physician in the analysis plan.

This is a good idea that we briefly discussed during the development of the study, but the number of primary health care centres and different physicians is too high. Moreover, physicians often only work temporarily at a specific health care centre and far from every patient see the same physician each time.

9. Please specify where and how you will get the information for the medical chart review and medication review.

Thank you for your comment. We realize that this procedure can probably differ between health care settings and deserves to be explained. All clinical pharmacist have electronic access to the medical charts (including both primary health care and hospital records) through their employment at Region Västerbotten, which is the government-funded regional health care provider. In the medical charts, information for the different activities mentioned are mainly gathered from physicians' notes, current

list of medications, laboratory records, and epicrises. For the sake of clarity, we have further described our access to medical chart review in more detail on page 7.

10. The planned subgroup analysis for heart failure and cognitive impairment makes sense as you expect a difference in effect of the intervention in those groups based on other studies. However, as this is a randomized study and the groups are even stratified for cognitive impairment, you do not expect a rate difference between the intervention and control arms and thus no adjustment of the regression model should be necessary.

We do agree that stratification and adjustment for relevant prognostic factors (in this case cognitive impairment) are both important means to increase the efficiency of the statistical analysis. However, our opinion is that one measure does not exclude the other, and it is important that the analysis reflect the restriction that the stratification imposes on the randomization. This approach is recommended in the guidelines regarding adjustment for covariates in clinical trials issued by the European Medicines Agency

(https://www.ema.europa.eu/en/documents/scientific-guideline/guidelineadjustment-baseline-covariates-clinical-trials_en.pdf, accessed on 21 February 2020). These recommendations promote that the stratification variable should only be excluded

from the primary analysis model if the stratification was solely conducted for administrative purposes. Ideally, we would have applied the same principle on heart failure, but restricted information regarding diagnoses before randomization has unfortunately made this option unfeasible.

Abstract

1. The intervention should be better described in the summary.

We agree that this kind of complex intervention needs to be sufficiently described and we have expanded our section regarding the intervention group protocol. We have also added an important clarification in the abstract about the actual intention of the intervention itself, which is to identify and manage drug-related problems. Due to the word count restriction, we could unfortunately not do a more detailed description in the abstract.

2. Point for the subgroup analysis applies to the abstract also.

Please, see our response to comment no. 10.

Reviewer 2:

Reviewer Name: Ulrich Jaehde

Institution and Country: Institute of Pharmacy, University of Bonn, Germany Competing interests or state: None declared

The manuscript describes the protocol for a randomised controlled trial investigating the effect of a clinical pharmacist intervention in elderly patients after discharge from the hospital. The intervention consists of three telephone calls and repeated medical chart reviews by the clinical pharmacist. As there is a strong need for specific interventions assuring patient safety in transitions of care, the study results will be of great interest for health care providers and quality managers striving for a safer medication process after discharge. The study is well recruiting. From September 2018 to November 2019 161 patients were randomised.

The manuscript is well written, the methods are state-of-the-art and the protocol is described clearly and presented in the context of the literature. The protocol addresses the items of the SPIRIT checklist where appropriate. Therefore, the manuscript is acceptable for publication after consideration of the following minor issues:

1. Is the time that the clinical pharmacist needs for the intervention documented? If yes, this should be mentioned as it will be an important information for potential future financing of the intervention.

As you describe, this information is highly relevant in order to assess the potential for implementing this type of intervention into routine care. Therefore, the clinical pharmacist thoroughly document the time spent on the different activities for each participant. This activity has now been mentioned in the revised version of the manuscript on page 7.

2. On page 9 the authors state that a medication review is conducted in the event of new medications or any documented drug-related symptom. This information should be added to the scheme in Figure 2.

We kindly appreciate your suggestion. We have added this information to Figure 2 so that it conforms with the description on page 7.

3. Page 10: It is not clear if the outcome measurements after 30 and 180 days are performed orally or using a questionnaire. If it is done by a phone interview is it after 30 days done in conjunction with the phone call which is part of the intervention?

Thank you for noticing this missing information. These outcome measurements are performed orally by phone. Unless the participant has been transferred to another ward prior to discharge, which in turn delays the intervention, these follow-ups are carried out in conjunction with the intervention phone calls. We have now made a clarification on page 8.

4. Page 11: The assessment of the causality of readmission is a major issue but remains obscure. The authors state that they will conduct a review of available options in order to find a valid and feasible classification tool. Since 16 months have passed after initiation of the study: has this review process been finished in the meantime? If yes, what is the result?

If not, please give more details on the available options.

Thank you for your comment. We have not found a method that we think can replace the original plan to summon a group of clinical experts in the form of physicians and pharmacists. We have now decided to use this method in order to be more transparent in our protocol. The blinded expert group of external health care professionals will judge the causality of each hospital readmission according to WHO-criteria. This has been specified on page 9.

5. Page 12: The qualitative part of the study should be explained in more detail, especially regarding the methodology.

Thank you, we understand that this section seemed a bit short on information. The qualitative part of the study has not yet begun and the interview guide is still under development. We have now expanded this section on page 10 with a more detailed description of the methodology.

6. Authors' contributions: The first author of the manuscript (assuming that he is one of the PhD students) is also one of the clinical pharmacists performing the intervention. Thus, he is in a double role and in fact evaluating his own performance. The authors should comment if this "personal union" can cause a major bias in the study results which should be considered when interpreting the results. The assumption is valid and this potential bias should not be neglected. It is possible that the element of self-evaluation adds an extra dimension to the potential risk of confirmation bias. However, we do not consider this to be a major issue since we have taken several steps to reduce the risk of bias as much as possible.

1. First of all, the assessment of drug-related readmissions is limited to health care professionals who are not otherwise involved in the project (page 6 and 9).

2. Data on admissions and other types of hospital visits are electronically recorded non-subjective clinical data with no room for misinterpretation. On the other hand, the questionnaires are more vulnerable to bias. We will be clear and transparent about this issue when we present and discuss our results.

3. The first author will most likely be involved in the statistical analysis and therefore we will also consult an external statistician in this process. This is also mentioned on page 6.

4. Regarding the qualitative aspect of the study, any interview about the intervention itself will be conducted by another person than the clinical pharmacist who performed that specific intervention. This is now added in the new section Limitations and risk of bias on page 12.

VERSION 2 – REVIEW

REVIEWER	Prof. Ulrich Jaehde, Ph.D. Institute of Pharmacy University of Bonn Germany
REVIEW RETURNED	14-Mar-2020
GENERAL COMMENTS	The authors have done a great job in addressing the reviewers' comments. The manuscript is now ready for publication.